# Optimal Vegetable Intake for Metabolic-Dysfunction-Associated Steatotic Liver Disease (MASLD) Prevention: Insights from a South Italian Cohort

**DOI:** 10.3390/nu17152477

**Published:** 2025-07-29

**Authors:** Maria Noemy Pastore, Caterina Bonfiglio, Rossella Tatoli, Rossella Donghia, Pasqua Letizia Pesole, Gianluigi Giannelli

**Affiliations:** 1Unit of Data Science, National Institute of Gastroenterology-IRCCS "Saverio de Bellis", Castellana Grotte, 70013 Bari, Italy; noemy.pastore@irccsdebellis.it (M.N.P.); rossella.tatoli@irccsdebellis.it (R.T.); rossella.donghia@irccsdebellis.it (R.D.); 2Core Facility Biobank, National Institute of Gastroenterology-IRCCS "Saverio de Bellis", Castellana Grotte, 70013 Bari, Italy; letizia.pesole@irccsdebellis.it; 3Scientific Direction, National Institute of Gastroenterology-IRCCS "Saverio de Bellis", Castellana Grotte, 70013 Bari, Italy; gianluigi.giannelli@irccsdebellis.it

**Keywords:** metabolic-dysfunction-associated steatotic liver disease (MASLD), color vegetable, mediterranean diet

## Abstract

(1) Background: Metabolic-dysfunction-associated steatotic liver disease (MASLD) is now the most prevalent chronic liver disease worldwide, posing a growing public health concern. While dietary improvements are key to prevention, the impact of different vegetable types remains unclear. This study focuses on the association between vegetable consumption and the risk of MASLD in a cohort of Southern Italy. (2) Methods: This research involved 1297 participants from the NUTRIHEP study, examining overall vegetable intake and classifying them into color subgroups to determine optimal quantity and variety for risk reduction. (3) Results: Daily consumption of approximately 325 g (two servings) of total vegetables significantly reduces the risk of MASLD (OR: 0.521; 95% CI: 0.317; 0.858). Among the subgroups, green vegetables were most protective at 35 g/day, while red and orange vegetables offered protection at 130 g/day. A higher intake of the other vegetable category, specifically onions, was associated with a reduced probability of MASLD (OR = 0.995; 95%CI: 0.989; 0.999). (4) Conclusions: These findings suggest a threshold effect, where moderate but regular consumption of specific vegetables offers maximal protection. Consuming excessive amounts may not enhance this benefit within this cohort. Cultural and regional dietary patterns should be considered when designing targeted nutritional interventions.

## 1. Introduction

Metabolic-dysfunction-associated steatotic liver disease (MASLD), previously named non-alcoholic fatty liver disease (NAFLD), is defined as the presence of excess triglyceride storage in the liver in the existence of one or more cardiometabolic risk factor such as obesity, type 2 diabetes, hypertension, or dyslipidemia and the absence of harmful alcohol intake [1].

MASLD is currently the most common chronic liver disease, with an estimated prevalence of 38% of the global adult population and around 13% of children and adolescents, becoming a major threat to public health due to its very high prevalence and related morbidity and mortality [2].

The pathogenesis of MASLD is complex and influenced by various factors, including metabolic, genetic, environmental, and lifestyle elements [3]. A key factor in this condition is dysregulated lipid metabolism, which leads to excessive lipid accumulation in the liver and hepatocyte damage. Furthermore, insulin resistance and hyperinsulinemia seem to have a huge relevance in amplifying lipogenesis and reducing lipid oxidation, which promote steatosis and metabolic stress [4]. Insulin resistance is often associated with obesity and type 2 diabetes, which represent the metabolic diseases with the strongest impact on the natural history and progression of MASLD [5].

Chronic inflammation and oxidative stress, often stemming from mitochondrial dysfunction and lipid peroxidation, promote cellular injury and fibrogenesis, driving progression to metabolic-dysfunction-associated steatohepatitis (MASH), characterized by hepatic inflammation and fibrosis, which may further lead to cirrhosis and hepatocellular carcinoma [4,6].

Genetic predispositions and epigenetic modifications also play significant roles in MASLD susceptibility and progression [7,8]. Environmental and lifestyle factors, including dietary habits and physical inactivity, significantly impact the development and progression of MASLD, underscoring the importance of holistic approaches to prevention and treatment [3].

The management of MASLD is multifaceted, involving lifestyle modifications, pharmacotherapy, and, in some cases, surgical interventions. Dietary changes, weight loss, physical exercise, and discouraging alcohol consumption are considered the most effective recommendations to prevent the development of MASLD and its complications [5].

A Mediterranean diet, the Dietary Approaches to Stop Hypertension (DASH) diet, low-carbohydrate and ketogenic diets, and intermittent fasting have shown benefits in reducing liver fat, improving insulin sensitivity, and mitigating inflammation [9,10,11,12]. All these dietary patterns emphasize the importance of incorporating a generous and regular intake of raw and cooked vegetables, highlighting their essential role in promoting overall health and well-being. On the contrary, diets rich in red and processed meat as well as ultra-processed foods (UPFs) are associated with a higher risk of developing MASLD and MASLD-related conditions due to high energy intake and saturated fat [13,14].

Diets high in red meat are contrasted with those rich in vegetables, which seem to have the most beneficial effect on hepatic steatosis due to their antioxidant and anti-inflammatory effects [11,15]. However, it is unclear whether all types of vegetables benefit MAFLD. Vegetables rich in flavonoids, carotenoids, and α-tocopherol have been shown to reduce the risk of MASLD and possibly effectively prevent liver fibrosis [16,17,18,19].

Recent studies have investigated the association between different types of vegetables and MASLD in other populations [18,20]. Consumption of dark green vegetables has been associated with reduced odds of MASLD, particularly among females and non-Hispanic white individuals [20]. Frequent consumption of soy products was significantly and negatively associated with the onset of MASLD in Japanese women [21]. However, there are not enough studies investigating this specific topic in the Mediterranean region, particularly in the southern Italian areas.

This study aims to assess how various types of vegetables influence the risk of MASLD in a Southern Italian cohort, and to determine the optimal quantity and variety of vegetable consumption necessary to reduce that risk.

## 2. Materials and Methods

### 2.1. Study Population

The NUTRIHEP study is a cohort initiated in 2005–2006, using a systematic random sample of attendees over 18 years from Putignano Primary Care Physicians’ list procedure [22].

We used data from General Practitioners’ (GP) registers instead of census records, as there were no notable differences in the age and gender distribution between the general population of Putignano and the data recorded in the GP registers. Between 2015 and 2018, all Nutrihep participants were invited for the initial follow-up. A total of 1426 participants responded, and they followed the same protocol as during the initial enrollment. All participants signed informed consent forms after receiving detailed information about the medical data to be studied. To address this, this study was designed as cross-sectional, focusing solely on the follow-up measurement. This study was approved by the Ethical Committee of the Minister of Health (DDG-CE-792/2014, on 14 February 2014).

### 2.2. Data Collection

During follow-up visits, the participants completed all assessments outlined in the protocol. Trained physicians and/or nutritionists interviewed them to collect data on sociodemographic details, health status, personal history, and lifestyle factors. This included a history of tobacco use, food intake, educational level [23], work profile [24], and marital status.

The participants’ weight and height were measured while wearing only underclothing and no shoes. We used an electronic balance (SECA©, Hamburg, Germany) to record weight to the nearest 1 kg, and a wall-mounted stadiometer (SECA©) to measure height to the nearest 1 cm. Blood pressure (BP) was measured following international guidelines [25,26], and the average of three measurements was calculated. The participants completed the EPIC food questionnaire independently to gather information about their eating habits [27,28]. The following blood measurements were taken: fasting serum glucose (FSG), fasting insulin, HbA1c, triglycerides, total cholesterol, LDL-C, HDL-C, AST, ALT, GGT, ferritin, and high-sensitivity C-reactive protein. Analyses were performed using the COBAS 8000 autoanalyzer (ROCHE Diagnostics SPA, Monza, Italy).

Insulin resistance was estimated using the Homeostasis Model Assessment of Insulin Resistance (HOMA-IR) [29], calculated by the following formula:HOMA-IR = FSG (mg/dL) × fasting Insulin (μIU/mL)/405

All subjects underwent standardized ultrasound exams using a Hitachi H21 Vision (Hitachi Medical Corporation, Tokyo, Japan). The liver parenchyma was examined with a 3.5 MHz transducer. A scoring system was used to semi-quantitatively assess hepatic fat content.

The degree of hepatic fat infiltration was determined based on liver echotexture, hepatic echo penetration, hepatic blood vessel clarity, and hepatic diaphragm differentiation in echo amplitude [30]. Appendix A illustrates the ultrasound board employed in this study to determine the steatosis grade.

The European Prospective Investigation into Cancer and Nutrition (EPIC) Food Frequency Questionnaire (FFQ) [31] was used to document habitual food intake at baseline. The participants self-completed the EPIC questionnaire and returned it after one week. Nutritionists validated all responses and uploaded the questionnaire into a custom online tool. Afterwards, the entered nutritional data was transformed into micro- and macro-nutrients.

### 2.3. Outcome Assessment

MASLD is defined by the presence of hepatic steatosis combined with at least one of the following cardiometabolic risk factors: (1) BMI over 25 kg/m^2^ or waist circumference exceeding 94 cm in men and 80 cm in women; (2) fasting serum glucose of 100 mg/dL or higher, 2 h post-load glucose of 140 mg/dL or higher, HbA1c of 5.7% or higher, or being on specific medication; (3) blood pressure of 130/85 mmHg or higher, or on specific medication; (4) plasma triglycerides at or above 150 mg/dL, or on specific medication; (5) plasma HDL cholesterol below 40 mg/dL in men and below 50 mg/dL in women, or on specific medication. Additionally, the MASLD definition continues to restrict alcohol intake (similar to NAFLD) in those with steatosis to an average daily intake of 20–50 g for women and 30–60 g for men [1].

Finally, other forms of liver disease coexisting with MASLD, such as MASLD + HCV, HBV, were ruled out to avoid altering the natural history of the disease [32] (Figure 1).

### 2.4. Exposure Variable

The EPIC questionnaire provided data on the frequencies and amounts of 30 vegetables, which we later categorized by color, as displayed in Table 1. Neither the type of cooking nor whether the vegetables were consumed raw or cooked was considered in this study.

### 2.5. Confounding Variables

Covariates were chosen based on previous research and both clinical and statistical judgment of their potential link to MASLD. After evaluating collinearity risks, we included demographic variables such as Age, Gender, Education, and Personal assessment of family income; behavioral factors like smoking, red wine consumption, daily energy intake, total vegetable intake regardless of exposure category, and food groups (fruits, legumes, cereals, fresh fish, olive oil, total meat, dairy products) excluding vegetables; and laboratory measures including AST/ALT, HOMA, and ɣgt.

To prevent collinearity issues, variables defining MASLD- such as BMI, waist circumference, fasting glucose, triglycerides, blood pressure, HDL cholesterol, and HbA1c- were excluded.

### 2.6. Statistical Analysis

The individual characteristics are reported as means and standard deviations (M ± SD) or medians and interquartile ranges for continuous variables and as frequencies and percentages (%) for categorical variables. We applied the Wilcoxon rank-sum test for comparing continuous variables between two groups. For categorical variables, we used the χ^2^ test to assess differences. Additionally, a logistic regression model was fitted to estimate odds ratios (ORs) and 95% confidence intervals (CIs), with MASLD as the outcome variable and vegetable intake (both continuous and categorical) as predictors.

Odds ratio (OR) is a measure of association between an exposure and an outcome.

An odds ratio of 1 indicates no association between the exposure and outcome. An odds ratio greater than 1 suggests that the exposure is associated with an increased likelihood of the outcome (a risk factor). An odds ratio of less than 1 suggests that the exposure is associated with a decreased likelihood of the outcome (a protective factor [33]).

The models were adjusted for the following variables: age, gender, smoking, education, daily kcal, γGT, AST/ALT, HOMA, intake of total vegetables devoid of exposure category, food groups (fruits, legumes, cereals, fresh fish, olive oil, total meat, dairy products) without vegetables, personal assessment of family income and red wine intake (g/day), and the estimated coefficients were transformed into odds ratio (OR).

Models in which the exposure variable was total daily vegetable consumption were adjusted for age, gender, smoking, education, daily kcal, γGT, AST/ALT, HOMA, Food groups (fruits, legumes, cereals, fresh fish, olive oil, total meat, dairy products) without vegetables, personal assessment of family income, and red wine intake (g/day).

Initially, confounding variables were chosen based on the existing literature. Then, the minimum absolute reduction and selection (LASSO) method was used to reduce the number of candidate predictors and highlight the most valuable ones for building the model [34]. When selecting variables as confounders for the model, those already included in the definition of MASLD, such as BMI, waist circumference, HDL cholesterol, triglycerides, fasting glucose, HbA1c, and blood pressure, were excluded. Additionally, the Variance Inflation Factor (VIF) was assessed to detect multicollinearity, and confounders with VIF > 5 were removed (Appendix A) [35].

We conducted a detailed sensitivity analysis to identify the optimal exposure threshold with the most significant protective effect on outcomes, where total vegetable intake was significantly related to an increased likelihood of MASLD. We calculated intake increments between the different vegetable groups to identify the daily intake category with the lowest odds ratio (OR) and statistical significance, based on daily intake.

Total vegetable intake was ranked in 25 g/day intervals from 150 (lower) to 400 (higher) g/day. The intake of green vegetables was ranked in 5 g/day intervals from 30 (lower) to 55 (upper) g/day. The intake of red and orange vegetables (g/day) was classified into 10 g/day intervals, ranging from 70 g/day (lower) to 170 g/day (upper).

A forest plot was created to compare the OR values from the model fitting, using Total Vegetable intake as a categorical variable.

The two-tailed probability level was set at 0.05 to test the null hypothesis of non-association.

The analyses were conducted with StataCorp 2025 Stata Statistical Software: Release 19 (College Station, TX, USA: StataCorp LLC.), while the forest plots were created using RStudio (“Mariposa Orchid” Release (ab7c1bc7, 2025-06-01)) and its packages “forestplot”.

## 3. Results

### 3.1. Participants’ Characteristics

Table 2 presents the main characteristics of the 1297 participants, classified according to the presence or absence of MASLD. The sample consisted of 48.50% with MASLD, with most of them being male (54.6%).

As shown in Table 2, the participants with MAFLD were older and mainly female. They also had a higher prevalence of hypertension and hyperlipidemia, along with greater BMI and weight: 30.28 (4.97) and 79.58 (14.73), respectively. Those with MASLD had a lower educational level compared to non-MASLD participants; 423 individuals had primary or secondary education as their highest level, while 214 had no MASLD. Among MASLD subjects, 59 were college graduates versus 119 in the non-MASLD group. Blood parameters were higher in MASLD patients, with statistically significant differences between groups. No significant differences in vegetable intake were observed; however, MASLD subjects consumed fewer green vegetables and more red/orange and other vegetables. There were no differences in Mediterranean Diet scores between groups, indicating similar adherence regardless of disease status. Likewise, macronutrient and micronutrient intake showed no significant variation between MASLD and non-MASLD groups (Appendix A).

In Table 3, we have broken down the vegetables by MASLD, demonstrating no statistically significant differences between those with MASLD and those without, except for the consumption of onions.

### 3.2. The Associations Between Different Kinds of Vegetable Intake and the Occurrence of MASLD

Table 4 presents the results of logistic regression models that examine the association between MASLD and total vegetable intake, presented as either a continuous or a categorical variable.

The results of the logistic regression model shown in Table 4 indicate that the category with the lowest OR was total vegetable consumption ≤325 (g/day) [OR 0.521 (95% CI 0.317; 0.858)] versus consumption >325 (g/day), after adjustment for covariates. This suggests that individuals with a maximum daily vegetable intake of 325 g had a 47.9% probability of not having MASLD, considering a model adjusted for age, gender, daily calories, ɣGT, AST/ALT, food groups without vegetables, red wine intake, and personal income assessment.

The odds ratio (OR) slightly above 1 (OR = 1.002; 95% CI: 1.001; 1.004, *p*-value = 0.007) indicates a positive association between daily vegetable intake and the likelihood of having MASLD. In simpler terms, this implies that individuals who consume more vegetables may be slightly more likely to develop MASLD.

The forest diagram (Figure 2) illustrates the OR values for each category of total vegetables, aiming to identify the one with the lowest risk value.

Multivariate logistic regression models were subsequently employed to verify the associations between MASLD and various types of vegetable intake (see Table 5).

We performed a detailed analysis to determine the optimal exposure cutoff with the most significant protective effect on the outcome (Table 5).

Green, red/orange vegetables were included in the models as both continuous and categorical variables, while other vegetables were considered as total continuous intake. The total intake and categories of vegetable consumption (g/day) with the lowest statistically significant odds ratios are presented in Table 5.

After adjustment for covariates, it was observed that the intake of the other vegetables group was negatively associated with MASLD (OR = 0.995, 95%CI: 0.989; 0.999, *p*-value = 0.047), while a higher intake of green vegetables and red/orange vegetables was significantly associated with increased odds of MASLD (OR = 1.005; 95% CI: 1.001; 1.009, *p*-value = 0.009, OR = 1.004; 95% CI: 1.001; 1.007, *p*-value = 0.004, respectively). In simpler terms, this implies that individuals who consume more green vegetables and red/orange vegetables may be slightly more likely to develop MASLD (see Table 5).

Table 5 shows the consumption categories for each group of vegetables studied. In the category “green vegetables”, the most protective intake was 35 g/day (OR = 0.616; 95% CI: 0.446; 0.851; *p*-value = 0.003). In other words, individuals consuming a daily intake of 35 g of “green vegetables” had a 39% reduced probability of not having MASLD compared to those with the highest consumption. However, eating up to 45 g of green vegetables daily can help protect against MASL, as this effect remains statistically significant.

In the “orange-red vegetables” category, it was an intake of 130 g per day that provided the greatest protection against MASLD (OR = 0.457; 95% CI: 0.274; 0.762; *p*-value 0.003). Thus, subjects consuming up to 130 g/day of red and/or orange vegetables had a 54.3% chance of not having MASLD compared to those with the highest consumption. Nevertheless, consumption of 150 g/day of red/orange vegetables still provides a statistically significant protective effect against MASLD.

## 4. Discussion

This research explored the impact of various vegetable types on the risk of developing MASLD among a cohort of 1297 Southern Italians from the NUTRIHEP study.

Our findings indicate that the greatest benefit was observed at an intake of approximately 325–350 g per day of total vegetables, equivalent to about two servings. This ideal daily amount would align with the national food-based dietary guidelines and the Mediterranean Diet, which suggests consuming at least two servings of vegetables per day [36,37]. Interestingly, in this cohort, consuming more than this amount did not provide any additional protection.

This indicates that, despite current scientific evidence consistently indicating that increased vegetable consumption is associated with a reduced risk of MASLD [11,15,18,20], certain factors may mitigate this protective effect. Along with following healthy dietary patterns and maintaining high overall diet quality, genetic variations may influence individual responses to dietary components [38]. A further study conducted on our cohort showed that a balanced diet, based on the principles of the Mediterranean Diet and rich in vegetables, combined with physical activity, can reduce the risk of MASLD and improve liver health in subjects with a specific genetic profile, such as FOXO3 TT [7]. This suggests that genetic variants could contribute to the development of MASLD, independent of vegetable intake. Additionally, various lifestyle factors, such as a lack of physical activity, may reduce the protective effects of vegetables on the liver [39]. Socioeconomic factors, medication use, and the presence of additional comorbidities could also have influenced the results of this study [3].

It is also worth noting that the population group investigated in this study moderately adheres to a Mediterranean Diet pattern, specifically using extra virgin olive oil as the primary seasoning, recognized for its antioxidant and anti-inflammatory effects. These dietary habits already serve as a protective factor against MASLD, which may explain why increasing vegetable consumption in this population might not necessarily reduce the odds of developing MASLD [40]. On the other hand, in different populations, including Chinese, Korean, and American adults with varying dietary habits, the increased consumption of vegetables seems to decrease the risk of NAFLD or MASLD [15,18,20]. These findings support the hypothesis that results may vary between populations, as previously hypothesized by Wang et al. [41].

When analyzing vegetables by color group, a similar trend emerged. Vegetables were grouped based on their pigments and bioactive compounds. For instance, the green group includes all vegetables rich in chlorophyll, which gives them their characteristic color, as well as compounds like β-Carotene, which confer antioxidant properties. For each color group, we identified a specific intake threshold at which the protective effect against MASLD was strongest. Red/orange vegetables, primarily represented by tomatoes and carrots, are most effective when eaten at a daily intake of 130 g, followed by green vegetables, especially leafy greens, lettuce, and cruciferous, at 35 g per day. In this cohort, exceeding these thresholds did not provide any additional protection. Conversely, for other vegetables, such as purple and white varieties—particularly onions, fennel, and celery—we did not identify a threshold.

Red/orange vegetables are rich in carotenoids, especially lycopene, which is abundant in tomatoes, a key ingredient in southern Italian cuisine. Lycopene has shown significant benefits for liver health and plays a crucial role in preventing and managing liver disease. It helps improve liver enzyme levels, reduces oxidative stress, lowers inflammation, and modulates gut microbiota—all of which are essential for preventing liver damage [16,42,43,44]. Lycopene supplementation has been shown to suppress hepatic inflammation by inhibiting the NF-κB/NLRP3 inflammasome pathway, leading to decreased expression of pro-inflammatory cytokines such as TNF-α and IL-6 [43]. Lycopene stands out as a promising agent in preventing MASLD due to its comprehensive benefits on liver health and metabolic regulation.

The relationship between green leafy vegetables and metabolic-dysfunction-associated liver disease has been examined in several studies [18,20]. Additionally, it is observed that including a green leafy vegetable component in diets may mitigate the detrimental impacts of other foods. For instance, adding a leafy vegetable portion could decrease the mortality risks associated with red meat for MASLD patients, and substituting one serving of starchy carbohydrates with green leafy vegetables may enhance NAFLD biomarkers and lessen fibrosis [45,46]. The beneficial effects of these vegetables are attributed to their antioxidant properties and bioactive compounds, including α-tocopherol, flavonoids, and carotenoids, particularly β-carotene [17,42,47].

β-Carotene, a provitamin A carotenoid found abundantly in vegetables like spinach and kale, acts as a powerful scavenger of free radicals, protecting cellular components from oxidative damage. It also modulates immune responses by attenuating inflammatory signaling pathways, such as NF-κB and MAPK, thereby reducing the production of pro-inflammatory cytokines [42,48]. α-Tocopherol, the most active form of vitamin E in humans, is abundantly found in green leafy vegetables, although in lower concentrations compared to oils and nuts. It is a potent antioxidant that has shown potential therapeutic effects on liver enzyme levels and histological features in MASLD patients, particularly in reducing oxidative stress and inflammation [47,49]. Additionally, green leafy vegetables, including lettuce, are high in inorganic nitrate, which may aid in treating and preventing fatty liver disease [50]. Inorganic nitrate can be converted into nitric oxide (NO) through a process involving symbiotic host bacteria. Nitric oxide has the potential to reduce oxidative stress and improve cardiometabolic functions [51].

Cruciferous vegetables, largely because of their component sulforaphane (SFN), protect against NAFLD by enhancing the gut barrier, modulating gut microbiota, reducing harmful bacteria, inhibiting the production of lipopolysaccharides, and decreasing inflammation. Additionally, SFN appears to improve insulin resistance, one of the key factors in the physiopathology of MASLD [52,53].

Finally, among the “White and other color vegetables” group, which has demonstrated a consistent protective effect, the most consumed vegetables are fennel and onions. This is likely due to their availability throughout much of the year, especially onions, which are widely eaten in this region. Interestingly, these vegetables seem to lower the risk of MASLD in this cohort, regardless of quantity. Onion has been shown to possess numerous pharmacological properties, attributable to its rich content of bioactive compounds such as quercetin, thiosulphinate, and phenolic acids [54]. One of the main protective effects is on the gastrointestinal system, resulting in stimulation of beneficial microorganism growth, as well as normalization of liver enzymes activities. A study conducted by Emamat et al. [55] showed that onion consumption can help in NAFLD management when combined with a healthy dietary pattern, and regular consumption of onion powder can prevent the development of NAFLD, even in the presence of other risk factors such as obesity and high energy, fat, and sugar intakes [56].

The dietary habits and traditions of the studied group may help explain the findings. In this region of Southern Italy, it is common to find local fruits and vegetables that are available according to the seasons. For example, green leafy vegetables are typically in season only at specific times of the year, which means their daily intake varies throughout the year. In contrast, red/orange vegetables, particularly tomatoes and carrots, along with onions, are available year-round and are widely used in many recipes, leading to a consistently higher daily intake. Eating locally sourced and in-season vegetables and fruits is a trend commonly observed in Mediterranean regions, along with other cultural elements [57]. This emphasizes the importance of adapting dietary interventions to geographic and cultural contexts as well [58].

Another factor that could explain the results of this study is the various cooking methods participants used to prepare different vegetables. Cooking methods can significantly affect the nutritional value of vegetables, especially concerning MASLD. While vegetables are typically beneficial due to their fiber, antioxidants, and anti-inflammatory properties, how they are prepared can change these benefits. For example, boiling can lead to the loss of certain polyphenols, which diminishes their protective effects against liver inflammation and fat accumulation. Frying, especially deep-frying, can reduce the active compounds’ content, and it can introduce harmful trans fats and oxidative compounds that may worsen insulin resistance and contribute to fatty liver disease. On the other hand, steaming keeps most nutrients intact and may even enhance the absorption of some antioxidants [59,60].

Improving nutrition literacy could be a cost-effective way to prevent chronic conditions [61]. Although many people are aware of healthy eating, they often struggle to put that knowledge into practice due to limited nutrition literacy. This is especially concerning in populations at risk for MASLD, where low vegetable intake is common. Factors like socioeconomic status, food access, and education level can prevent understanding and using nutrition information effectively [62]. Increasing nutrition literacy may help these individuals make healthier food choices, such as including the right amount and type of vegetables in their diets, and understand the importance of these choices for liver health.

The findings from this study align with the recommendations of the Mediterranean dietary pattern, which has been shown to help prevent or delay hepatic steatosis [63]. However, while the classic Mediterranean Diet suggests at least two servings of a variety of raw and/or cooked vegetables, our study indicates that consuming more than two servings does not provide additional benefits against MASLD. Additionally, in this study, we aimed to identify specific types of vegetables within the Mediterranean dietary pattern to better adapt to the dietary habits, preferences, and availability specific to different regions of the Mediterranean. Our findings suggest that a moderate intake of green leafy vegetables and cruciferous vegetables, along with a good consumption of tomatoes, carrots, and regular inclusion of onions, is beneficial against MASLD.

### Strengths and Limitations

One of the key strengths of this study is its large sample size, which boosts the statistical power and enhances the reliability of the findings. Additionally, this research is novel as it is the first to investigate the effects of different types of vegetables on MASLD in a population from Southern Italy. This regional focus provides valuable insight into dietary patterns specific to the Mediterranean context, potentially informing culturally tailored nutritional recommendations. However, several limitations should be acknowledged, such as the observational design, which limits causal inference. In addition, another potential limitation is related to self-reporting of diet. Furthermore, the EPIC FFQ is an international questionnaire that does not take into account typical local foods, which results in several regionally consumed vegetables being omitted. This may lead to an underestimation or overestimation of some quantities. Finally, our study did not include a measure of physical activity, which represents a potentially serious limitation given the results of previous research linking physical activity with MASLD [64,65]. These limitations should be taken into account when interpreting the results. The residual confounding was consistent across all models analyzed, and the differing environmental impacts could partly explain the inconsistencies with other studies [66].

## 5. Conclusions

In conclusion, the results of this study suggest that consuming approximately two servings of vegetables per day may help reduce the risk of MASLD in a southern Italian population. Notably, white vegetables, such as onions, play a significant protective role against MASLD, regardless of the quantity consumed. In contrast, increasing the intake of other types of vegetables may not necessarily offer additional benefits in this population. Therefore, these findings indicate that an effective dietary intervention for individuals at risk of MASLD should focus on the specific types and portions of vegetables to be consumed while also taking geographic and cultural factors into account. Encouraging adequate vegetable consumption may be important in preventing liver disease in this small geographic population. Further research is needed to establish optimal intake levels and to understand the underlying mechanisms of action.

## Figures and Tables

**Figure 1 nutrients-17-02477-f001:**
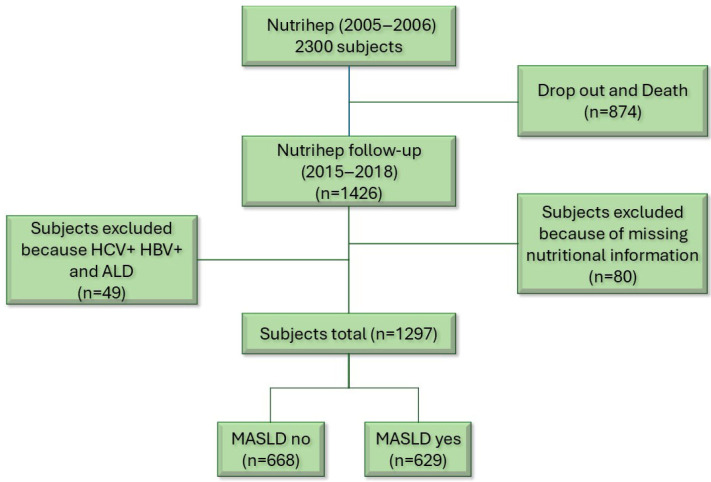
Flow chart.

**Figure 2 nutrients-17-02477-f002:**
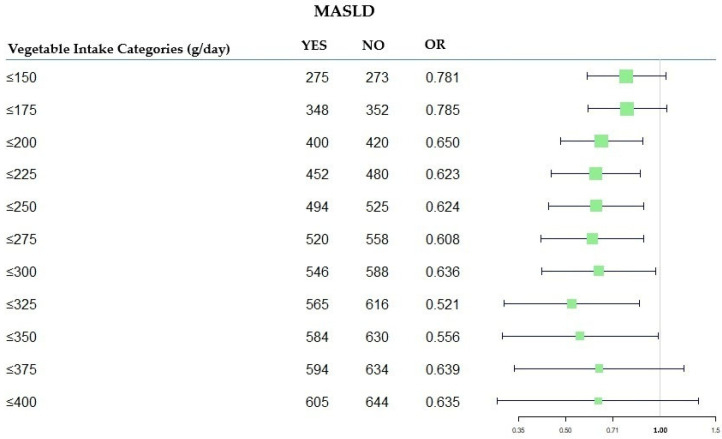
Forest plot of OR values in the MASLD by total vegetable categories (g/day). MASLD: Metabolic-Dysfunction-Associated Steatotic Liver Disease; OR: Odds Ratio.

**Table 1 nutrients-17-02477-t001:** Vegetables categorized by color.

** *Color group* **
**Green vegetables:** cruciferous (cabbage, broccoli, Brussel sprouts, turnips, kale, cauliflower), leafy greens (spinach, Swiss chard, chicory), leafy salads, lettuce, artichokes, green beans, zucchini
**Red/orange vegetables:** tomatoes, carrots, red beets
**White and other color vegetables:** eggplant, mushrooms, onions, leeks, fennel, celery, peppers, soy sprouts

**Table 2 nutrients-17-02477-t002:** Characteristics of participants by MASLD Nutrihep Study, Putignano (BA), Italy 2015–2018.

Variables ^a^		MASLD		
	Whole Sample ^b^	No	Yes	*p-value * ^c^
N (%)	1297	668 (51.50)	629 (48.50)	
**Exposure variables**				
Total vegetables (g/day)	183.17 (102.90)	185.85 (101.67)	180.33 (104.20)	0.33
Green vegetables (g/day)	53.85 (39.69)	54.31 (40.29)	53.38 (39.07)	0.67
Red and orange vegetables (g/day)	61.65 (52.69)	61.27 (52.82)	62.05 (52.59)	0.79
White and other color vegetables (g/day)	46.53 (29.93)	47.55 (29.16)	45.45 (30.72)	0.21
**Demographic and lifestyle characteristics**				
Age (years)	54.33 (14.34)	49.24 (13.80)	59.74 (12.86)	<0.001
Gender (%)				
Female	744 (57.4)	417 (56.0)	327 (44.0%)	<0.001
Male	553 (42.6)	251 (45.4)	302 (54.6%)	
Fruits (g/day)	409.17 (238.15)	395.87 (228.02)	423.29 (247.86)	0.038
Legumes (g/day)	33.89 (29.06)	33.20 (29.07)	34.63 (29.05)	0.37
Cereals (g/day)	219.22 (119.29)	223.85 (117.30)	214.31 (121.26)	0.15
Fish (g/day)	39.09 (26.04)	40.06 (25.44)	38.05 (26.64)	0.16
Olive oil (g/day)	18.17 (10.85)	18.26 (10.97)	18.08 (10.72)	0.76
Total meat (g/day)	93.54 (53.79)	95.36 (54.56)	91.60 (52.94)	0.21
rMED	8.04 (2.55)	7.91 (2.54)	8.18 (2.56)	0.05
rMED score (%)				
Low	365 (28.1%)	196 (53.7%)	169 (46.3%)	0.46
Moderate	705 (54.4%)	362 (51.3%)	343 (48.7%)	
High	227 (17.5%)	110 (48.5%)	117 (51.5%)	
Alcohol intake (g/day)	10.58 (12.72)	10.74 (13.41)	10.42 (11.96)	0.66
Wine intake (ml/day)	67.18 (174.36)	56.88 (214.44)	78.13 (116.89)	0.028
Kcal (day)	2056.26 (750.22)	2100.33 (724.88)	2009.46 (774.05)	0.029
Smoker (%)				
Never/former	1137 (87.7)	587 (51.6)	550 (48.4)	0.87
Current	159 (12.3)	81 (50.9)	78 (49.1)	
Marital Status (%)				
Single	181 (14.0)	115 (63.5)	66 (36.5)	<0.001
Married or living together	1034 (79.7)	519 (50.2)	515 (49.8)	
Separated or divorced	28 (2.2)	20 (71.4)	8 (28.6)	
Widow/er	54 (4.2)	14 (25.9)	40 (74.1)	
Education (%)				
Primary school	282 (21.8)	71 (25.2)	211 (74.8)	<0.001
Secondary school	383 (29.5)	171 (44.6)	212 (55.5)	
High School	460 (35.5)	307 (66.7)	153 (33.3)	
Graduate	172 (13.3)	119 (69.2)	53 (30.8)	
Work (%)				
Managers and professionals	102 (7.9)	57 (55.9)	45 (44.1)	<0.001
Craft, agricultural, and sales workers	469 (36.2)	285 (60.8)	184 (39.2)	
Elementary occupations	185 (14.1)	93 (50.3)	92 (49.7)	
Housewife	141 (10.9)	74 (52.5)	67 (47.5)	
Pensioner	325 (25.1)	110 (33.8)	215 (66.2)	
Unemployed	75 (5.8)	49 (65.3)	26 (34.7)	
Family income assessment (%)				
Insufficient	27 (2.1)	10 (37.0)	17 (63.0)	0.025
Just sufficient	167 (12.9)	81 (48.5)	86 (51.5)	
Sufficient	1019 (78.6)	521 (51.1)	498 (48.9)	
More than sufficient	64 (4.9)	44 (68.8)	20 (31.2)	
Good	20 (1.5)	12 (60.0)	8 (40.0)	
**Anthropometric and clinical parameters**				
BMI (kg/m^2^)	27.58 (5.05)	25.04 (3.59)	30.28 (4.97)	<0.001
Weight (kg)	72.93 (14.87)	66.66 (12.02)	79.58 (14.73)	<0.001
Waist (cm)	90.45 (13.46)	83.04 (10.38)	98.32 (11.79)	<0.001
SBP (mmHg)	120.93 (15.81)	115.64 (15.35)	126.52 (14.30)	<0.001
DBP (mmHg)	77.68 (8.00)	75.69 (7.88)	79.78 (7.58)	<0.001
Hypertension (%)				
No	847 (68.8)	517 (61.0)	330 (39.0)	<0.001
Yes	385 (31.2)	115 (29.9)	270 (70.1)	
Dyslipidemia (%)				
No	1047 (85.1)	561 (53.6)	486 (46.4)	<0.001
Yes	184 (14.9)	71 (38.6)	113 (61.4)	
Diabetes (%)				
No	1148 (93.2)	620 (54.0)	528 (46.0)	<0.001
Yes	84 (6.8)	12 (14.3)	72 (85.7)	
**Blood Test**				
HbA1c (mmol/mol)	38.07 (6.87)	36.59 (5.05)	39.64 (8.09)	<0.001
HOMA	1.89 (1.88)	1.33 (0.90)	2.43 (2.38)	<0.001
ALT (U/L)	22.20 (16.21)	19.70 (8.27)	24.86 (21.37)	<0.001
ɣGT (U/L)	17.58 (13.46)	14,80 (7,67)	20,54 (17,16)	<0.001
AST (U/L)	21.74 (10.87)	20,70 (5,94)	22,85 (14,29)	<0.001
TG (mg/dL)	98.41 (69.23)	80.73 (58.55)	117.22 (74.60)	<0.001
C-reactive protein (mg/dL)	0.26 (0.55)	0.21 (0.52)	0.31 (0.58)	<0.001
TC (mg/dL)	191.35 (35.36)	188.90 (33.06)	193.96 (37.50)	0.010
HDL (mg/dL)	50.79 (12.59)	53.18 (12.80)	48.24 (11.85)	<0.001
Glucose (mg/dL)	95.34 (17.34)	90.13 (10.54)	100.89 (21.06)	<0.001
ALP (U/L)	52.98 (16.10)	50.10 (15.56)	56.04 (16.11)	<0.001

^a^ As means and standard deviations. ^b^ Percentages calculated for the column. Otherwise, percentages are calculated for the row. ^c^ Wilcoxon rank-sum tests for continuous variables to compare two groups, and the χ^2^ test for categorical variables. MASLD: Metabolic-Dysfunction-Associated Steatotic Liver Disease; rMED: Relative Mediterranean Diet; BMI: Body Mass Index; SBP: Systolic Blood Pressure; DBP: Diastolic Blood Pressure; HbA1c: Glycosylated Hemoglobin; HOMA: Homeostasis Model Assessment; ALT: Alanine Amino Transferase; ɣGT: Gamma Glutamyl Transferase; AST: Aspartate Amino transferase; TG: Triglycerides; TC: Total Cholesterol; HDL-C: High-Density Lipoprotein Cholesterol; ALP: Alkaline Phosphatase Level.

**Table 3 nutrients-17-02477-t003:** Intake of different types of vegetables broken down by MASLD.

Variables		MASLD		
	Whole Sample	No	Yes	*p-value * ^a^
	Mean (SD)	Mean (SD)	Mean (SD)	
Green vegetables (g/day)				
Artichokes	3.31 (4.19)	3.32 (4.14)	3.30 (4.25)	0.94
Cruciferous ^b^	7.87 (7.96)	7.60 (7.41)	8.16 (8.50)	0.21
Green leafy vegetables ^c^	12.42 (15.49)	12.79 (15.76)	12.03 (15.19)	0.38
Lettuce	16.24 (19.96)	16.18 (22.23)	16.31 (17.24)	0.90
Zucchini	9.30 (10.39)	9.77 (10.49)	8.79 (10.27)	0.090
Green beans	4.71 (6.08)	4.65 (5.91)	4.78 (6.26)	0.70
Red/orange vegetables (g/day):				
Tomatoes	46.33 (47.02)	45.60 (48.32)	47.10 (45.63)	0.57
Carrots	13.87 (15.61)	14.14 (14.50)	13.59 (16.71)	0.53
Red beets	1.45 (2.46)	1.52 (2.26)	1.36 (2.65)	0.23
White and other color vegetables (g/day)				
Peppers	2.52 (3.51)	2.53 (3.42)	2.52 (3.62)	0.97
Mushrooms	4.32 (5.67)	4.54 (6.78)	4.09 (4.17)	0.15
Onions	11.74 (14.02)	11.02 (14.06)	12.51 (13.95)	0.050
Fennel	15.45 (13.79)	15.87 (13.24)	15.00 (14.36)	0.26
Eggplant	6.10 (6.86)	6.35 (6.80)	5.83 (6.92)	0.17
Soy sprouts	0.16 (0.82)	0.17 (0.85)	0.14 (0.79)	0.52
Celery	8.85 (9.86)	9.01 (9.85)	8.68 (9.88)	0.55

^a^ Wilcoxon rank-sum test for continuous variables. ^b^ Cruciferous: Cabbage, Broccoli, Brussel Sprouts, Turnips, Kale, Cauliflower. ^c^ Green Leafy Vegetables: Spinach, Swiss Chard, Chicory.

**Table 4 nutrients-17-02477-t004:** Logistic regression analysis of the association between total vegetable intake and MASLD. Daily vegetable consumption was expressed both continuously and categorically.

	OR ^a^	*p-Value*	95% CI
Total Vegetables (g/day)			
Categories:			
>150 vs. ≤150	0.781	0.096	0.584; 1.045
>175 vs. ≤175	0.785	0.101	0.588; 1.048
>200 vs. ≤200	0.650	0.005	0.480; 0.881
>225 vs. ≤225	0.623	0.004	0.450; 0.863
>250 vs. ≤250	0.624	0.008	0.440; 0.886
>275 vs. ≤275	0.608	0.010	0.416; 0.888
>300 vs. ≤300	0.636	0.035	0.418; 0.968
>325 vs. ≤325	0.521	0.010	0.317; 0.858
>350 vs. ≤350	0.556	0.046	0.313; 0.989
>375 vs. ≤375	0.639	0.161	0.342; 1.194
>400 vs. ≤400	0.635	0.228	0.303; 1.329
Total intake	1.002	0.014	1.000; 1.003

^a^ No MASLD reference categories. Models adjusted for: age, gender, smoking, education, daily kcal, γGT, AST/ALT, Food groups without vegetables, personal assessment of family income, and red wine intake (g/day). MASLD: Metabolic-Dysfunction-Associated Steatotic Liver Disease; AST: Aspartate Amino Transferase; ALT: Alanine Amino Transferase; ɣGT: Gamma Glutamyl Transferase.

**Table 5 nutrients-17-02477-t005:** Logistic regression analysis of the association between different kinds of vegetable intake and MASLD. Daily vegetable consumption was expressed both continuously and categorically.

	OR ^a^	*p-Value*	95% CI
Green vegetables (g/day)			
Categories:			
>30 vs. ≤30	0.670	0.019	0.480; 0.935
>35 vs. ≤35	0.616	0.003	0.446; 0.851
>40 vs. ≤40	0.656	0.009	0.478; 0.900
>45 vs. ≤45	0.704	0.029	0.514; 0.964
>50 vs. ≤50	0.734	0.056	0.534; 1.008
>55 vs. ≤55	0.813	0.212	0.588; 1.125
Total intake	1.005	0.009	1.001; 1.009
Red and orange vegetables (g/day)			
Categories			
>70 vs. ≤70	0.818	0.209	0.598; 1.119
>80 vs. ≤80	0.612	0.004	0.440; 0.851
>90 vs. ≤90	0.511	0.000	0.356; 0.733
>100 vs. ≤100	0.511	0.000	0.356; 0.733
>110 vs. ≤110	0.514	0.002	0.337; 0.784
>120 vs. ≤120	0.477	0.002	0.298; 0.762
>130 vs. ≤130	0.457	0.003	0.274; 0.762
>140 vs. ≤140	0.481	0.008	0.280; 0.829
>150 vs. ≤150	0.504	0.023	0.280; 0.910
>160 vs. ≤160	0.532	0.060	0.275; 1.027
>170 vs. ≤170	0.633	0.214	0.307; 1.303
Total intake	1.004	0.004	1.001; 1.007
White and other color vegetables (g/day)			
Total intake	0.995	0.047	0.989; 0.999

^a^ No MASLD categories. Models adjusted for: age, gender, smoking, education, daily kcal, γGT, AST/ALT, food groups without vegetables, personal assessment of family income and red wine intake (g/day). Legend: OR: Odds Ratio; MASLD: Metabolic-Dysfunction-Associated Steatotic Liver Disease; AST: Aspartate Aminotransferase; ALT: Alanine Aminotransferase; ɣGT: Gamma Glutamyl Transferase.

## Data Availability

The original data presented in this study are openly available in FigShare at https://doi.org/10.6084/m9.figshare.29390003.

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
