# Peer review of "Optimal Vegetable Intake for Metabolic-Dysfunction-Associated Steatotic Liver Disease (MASLD) Prevention: Insights from a South Italian Cohort"

_nutrients, 2025, doi:10.3390/nu17152477_

Round 1

Reviewer 1 Report

Comments and Suggestions for Authors

This study discusses the association between vegetable intake and MAFLD. It was designed well, and most of the methodology was also reasonable. However, several points need to be clarified before publication. 

Major

  1. The more conservative title should be considered for a cross-sectional study. The current prevention is inclined towards the prospective study.
  2. Abstract
    1. In addition to reporting OR, 95% CI is more informative
    2. Reconsider the interpretation for "Higher consumption did not confer additional benefit" as OR is very small: 1.002 with 95% CI: 1-1.003. This value is difficult to interpret as clinically significant, even though the P-value is <0.05. 
  3. Introduction: it will be better to emphasize the current lack evidences to address the unique of this study
  4. Line 103-104: The study design was therefore cross-sectional, considering only the follow-up measurements. Can you explain what does "conisdering only the folloiw-up measurements" mean in this cross-sectional study.
  5. Where is Figure S1?
  6. How many people perform the abdominal ultrasound? Do you have any internal validity?
  7. Figure 1: Are the values of the EPIC questionnaire in this study from 2015-2018 versus 2005-2006? It is not clear. Please clarify this question to understand better the study design (cross-sectional vs. an observational, prospective study?)
  8. Line 248-249 (Table S2): there are several variables showing P<0.05. But the author says no significant differences were observed in macronutrients and micronutrients between the MASLD and non-MASLD groups. Would you please help me interpret these results?
  9. Discussion: 
    1. I have pointed out in the abstract that the more vegetalbe consuming has a higher risk of MAFLD using OR: 1.002, 95% CI: 1.000-1.003, P-value: 0.014, which is not persuasive to me. Especially, the author also explained the genetic, or the protective Mediterranean Diet pattern may play some protective roles. Would you reconsider interpreting this result and discussion?
    2. If genetics and the Mediterranean Diet affect this study population, can the author provide other literature supporting the geographic variation/differences? Otherwise, the current inference seems arbitrary, and the evidence is insufficient. 

Minor

  1. Outcome assesment: why HBV is not in the excluded list?
  2. Line 186: Why is the Wilcoxon rank-sum test used for continuous variables instead of the independent t-test?
  3. Line 191-194: The Author can describe this more clearly by avoiding the "exposure group" or negative outcome. 
  4. Line P281-282: The numbers are not compatible with those in Table 4. 

Author Response

Comment 1: The more conservative title should be considered for a cross-sectional study. The current prevention is inclined towards the prospective study.

Response 1:

Comment 2: Abstract

    1. In addition to reporting OR, 95% CI is more informative
    2. Reconsider the interpretation for "Higher consumption did not confer additional benefit" as OR is very small: 1.002 with 95% CI: 1-1.003. This value is difficult to interpret as clinically significant, even though the P-value is <0.05. 

Response 2:

Comment 3: Introduction: it will be better to emphasize the current lack evidences to address the unique of this study

Response 3: Thank you for your comment. We have emphasized this concept in the introduction (line 84-86).

Comment 4: Line 103-104: The study design was therefore cross-sectional, considering only the follow-up measurements. Can you explain what "considering only the follow-up measurements" means in this cross-sectional study?

Response 4:Thank you for your comment. In this study we wanted to use data from the first follow-up of Nutriep, recruited in the period 2015-2018, because it was more recent data, compared to the 2005-2006 data, which we use in mortality studies. We were also not interested in comparing periods; we wanted to focus on more recent data. The same cohort, the same data and a similar statistical analysis can be found in our Antioxidants article “Exploratory Role of Flavonoids on Metabolic Dysfunction-Associated Steatotic Liver Disease (MASLD) in a South Italian Cohort; Antioxidants 2024 Oct 24;13(11):1286”.

Comment 5: Where is Figure S1?

Response 5: Through sheer distraction, it was not included in the supplementary material. We will do so in the paper update after the revisions are made.

Comment 6: How many people perform the abdominal ultrasound? Do you have any internal validity?

Response 6: The ultrasound was performed by two radiology technicians from our institute using an ultrasound card that was validated in the following study: Metab Syndr Relat Disord. Ultrasound evaluation and correlates of fatty liver disease: a population study in a Mediterranean area. 2013 Oct;11(5):349-58. This card has been in use at our Institute since 2006 although it was validated later. We will add this sheet in the supplementary material.

Comment 7: Figure 1: Are the values of the EPIC questionnaire in this study from 2015-2018 versus 2005-2006? It is not clear. Please clarify this question to understand better the study design (cross-sectional vs. an observational, prospective study?)

Response 7: As a cross-sectional study, the analysis focused on the data collected during the follow-up period.The nutritional values used in this work refer to the EPIC questionnaire that the subjects completed between 2015 and 2018, i.e., during the first follow-up. The reference to 2005-2006 is only included because this was the two-year period when the Nutriep cohort was established. We used the flowchart from a previously published paper, as the cohort and the study on nutrition and MASLD remained the same. Following your fair observations, we have reworked and revised Figure 1.

Comment 8: Line 248-249 (Table S2): there are several variables showing P<0.05. But the author says no significant differences were observed in macronutrients and micronutrients between the MASLD and non-MASLD groups. Would you please help me interpret these results?

Response 8: Thank you for your observation. We will reformulate better, highlighting those that are statistically significant.

Comment 9: Discussion: 

  • I have pointed out in the abstract that the more vegetalbe consuming has a higher risk of MAFLD using OR: 1.002, 95% CI: 1.000-1.003, P-value: 0.014, which is not persuasive to me. Especially, the author also explained the genetic, or the protective Mediterranean Diet pattern may play some protective roles. Would you reconsider interpreting this result and discussion?

Response 9.1: Thank you for pointing this out. We agree with your comment. We have accordingly modified the abstract and adjusted our discussion and conclusion as suggested by another reviewer as well. We clarified that, based on our results from this cohort, a higher intake of vegetables may not offer additional benefits in reducing the risk of MASLD. This does not imply that consuming more vegetables is harmful; rather, it suggests that eating two servings per day may be equally protective.

  • If genetics and the Mediterranean Diet affect this study population, can the author provide other literature supporting the geographic variation/differences? Otherwise, the current inference seems arbitrary, and the evidence is insufficient. 

Response 9.2: Lines 337-340

Minor

  1. Outcome assesment: why HBV is not in the excluded list?

Thank you for your observation. They fall within the 49 of the flow-chart. We will rectify for greater understanding.

  1. Line 186: Why is the Wilcoxon rank-sum test used for continuous variables instead of the independent t-test?
  2. The data in the descriptive tables were not always normally distributed, as required for the t-test, so we found it convenient to use a non-parametric test such as the Wilcoxon rank-sum test.
  3. Line 191-194: The Author can describe this more clearly by avoiding the "exposure group" or negative outcome. 

Thank you for your observation. I rephrased the sentence, using from a Statistics book, the definition of OR. I hope it is clearer expressed in this way.

  1. Line P281-282: The numbers are not compatible with those in Table 4. 

Thank you for your comment, but if the sentence is as follows, the exact values of the analysis described in Table 4 were reported: “The results of the logistic regression model shown in Table 4 indicate that the category with the lowest OR was total vegetable consumption ≤325 (g/day) [OR 0.521 (95% CI 0.317; 0.858)] after adjustment for covariates”.

>325 vs ≤325

0.521

0.010

0.317; 0.858

The probability of not having MASLD, in our study, when consuming a maximum of 325 g per day is given by (1-OR)*100, which is 47.1%.The comparison for the logistic analysis tables was made, with the highest consumption being taken as a reference and then compared with a lower consumption. The comparisons between the intake categories yielded reliable statistical results as long as the divisions within the intake categories were balanced in terms of numerosity. We have assumed that you are referring to this.

Reviewer 2 Report

Comments and Suggestions for Authors

The manuscript titled ‘  Optimal Vegetable Intake for Metabolic Dysfunction-Associ- ated Steatotic Liver Disease (MASLD) Prevention: Insights from a South Italian Cohort’ investigated the association between vegetable consumption and the risk of MASLDIt found out  that higher intake of vegetables is associated with a reduced likelihood of MASLD. Even though the authors have shown good efforts to provide evidences confirming the beneficial effects of vegetables in MASLD, this manuscript may require minor revisions before its publication. Some specific comments are provided below.

1. The rational behind classifying the vegetables by color should be explained from the perspective of qualityative and quantitative contents of bioactive components. Perhaps in teh discussions sction

2.  The conclusions statiting exceess vegetable intake thresholds can not provide any additional protection could be misleading considering that the authors did not conduct the study in a way to evaluate specific parameteres in a dose dependent response. May be citing previous literature on this issue is helpful

3. To avoid overexaggeration of the results in thsi study, authors can briefly add discussions on factors that may affect effects in the cohort such as physical activity, medication use, pathophysiological conditions e.g., diabetes, obesity, socioeconomic factors etc

4. In the conclusion section, authors can add what the results in this study can contribute public health or clinical practice

Comments on the Quality of English Language

-Longer sentences should be reduced into more readable and manageable ones 

-The names of some bioactive compounds such as carotenoids are redundant

Author Response

Comment 1 : The rationale behind classifying the vegetables by color should be explained from the perspective of qualitative and quantitative contents of bioactive components. Perhaps in the discussion section

Response 1:

Thank you for bringing this to our attention; we agree with your point. We have briefly explained this point in the discussions (lines 356-359). In the paper, vegetables were categorized based on their pigments and levels of other bioactive compounds, but we focused only on those most closely associated with liver health. Also, for statistical analysis purposes, we grouped vegetables of various colors, such as brown, purple, and undefined colors, together with the category called “White and other colour vegetables”. For some vegetables, for instance, peppers, the EPIC FFQ did not specify their classification, so we included them under other colors to avoid confusion.

Comment 2: The conclusions stating exceess vegetable intake thresholds can not provide any additional protection could be misleading, considering that the authors did not conduct the study in a way to evaluate specific parameters in a dose-dependent response.  Citing previous literature on this issue is helpful

Response 2:

Thank you for pointing this out. We have slightly changed the sentence, highlighting that a higher intake of vegetables might not provide additional benefits, meaning that it would not be negative, but simply might not be more protective than eating the two servings per day. Also, we have stated that this is a result found in this specific area of southern Italy, so it cannot be generalized. (lines 477-478)

Comment 3:

To avoid overexaggeration of the results in this study, authors can briefly add discussions on factors that may affect effects in the cohort, such as physical activity, medication use, pathophysiological conditions e.g., diabetes, obesity, socioeconomic factors etc

Response 3:

Thank you for your comment. We have emphasized this concept in the discussion (lines 342-345).

Comment 4:

In the conclusion section, authors can add what the results in this study can contribute public health or clinical practice

Response 4:

We agree with this comment. We have added a line to reinforce this concept. (lines 481-483)

Round 2

Reviewer 1 Report

Comments and Suggestions for Authors

All the questions were answered appropriately.